# SLAM-based augmented reality for the assessment of short-term spatial memory. A comparative study of visual versus tactile stimuli

**Francisco Munoz-Montoya**[1], **M.-Carmen Juan**[1]\*, **Magdalena Mendez-Lopez**[2], **Ramon Molla**[1], **Francisco Abad**[1], **Camino Fidalgo**[2]

**1** Instituto Universitario de Automática e Informática Industrial, Universitat Politècnica de València, Valencia, Spain, **2** Departamento de Psicología y Sociología, IIS Aragón, Universidad de Zaragoza, Teruel, Spain

\* mcarmen@dsic.upv.es

**Data Availability Statement:** All relevant data will be within the manuscript and its Supporting Information files.

## Abstract

The assessment of human spatial short-term memory has mainly been performed using visual stimuli and less frequently using auditory stimuli. This paper presents a framework for the development of SLAM-based Augmented Reality applications for the assessment of spatial memory. An AR mobile application was developed for this type of assessment involving visual and tactile stimuli by using our framework. The task to be carried out with the AR application is divided into two phases: 1) a learning phase, in which participants physically walk around a room and have to remember the location of simple geometrical shapes; and 2) an evaluation phase, in which the participants are asked to recall the location of the shapes. A study for comparing the performance outcomes using visual and tactile stimuli was carried out. Fifty-three participants performed the task using the two conditions (Tactile vs Visual), but with more than two months of difference (within-subject design). The number of shapes placed correctly was similar for both conditions. However, the group that used the tactile stimulus spent significantly more time completing the task and required significantly more attempts. The performance outcomes were independent of gender. Some significant correlations among variables related to the performance outcomes and other tests were found. The following significant correlations among variables related to the performance outcomes using visual stimuli and the participants' subjective variables were also found: 1) the greater the number of correctly placed shapes, the greater the perceived competence; 2) the more attempts required, the less the perceived competence. We also found that perceived enjoyment was higher when a higher sense of presence was induced. Our results suggest that tactile stimuli are valid stimuli to exploit for the assessment of the ability to memorize spatial-tactile associations, but that the ability to memorize spatial-visual associations is dominant. Our results also show that gender does not affect these types of memory tasks.

**Funding:** This work was funded mainly by FEDER/ Ministerio de Ciencia e Innovación – Agencia Estatal de Investigación/AR3Senses (TIN2017-87044-R); other support was received from the Generalitat Valenciana/Fondo Social Europeo (ACIF/2019/031); the Gobierno de Aragón (research group S31_20D) and FEDER 2020-2022 "Construyendo Europa desde Aragón". The funders had no role in study design, data collection and analysis, decision to publish, or preparation of the manuscript.

**Competing interests:** The authors have declared that no competing interests exist.

# 1. Introduction

People have the ability to store and remember representations of spatial stimuli for short periods of time. This ability is known as spatial short-term memory [1]. From neurobiological and cognitive perspectives, most of the information that humans explicitly store in spatial memory comes from the visual and auditory modalities [2–5]. However, the human brain has the ability to store information from all sensory modalities [6]. One of these sensory modalities is the sense of touch, which even though it is not the most studied sense, it is certainly a sense to exploit. Spatial memory is used to store and remember the route to find a previously visited place or to find where we have left our belongings (e.g., keys or glasses) [7]. This type of memory is involved in everyday tasks and allows navigation and solving spatial tasks. Thus, impairments in spatial memory have serious consequences in daily life. Different memory-related help tools can be used for assessment as well as training. For assessment, these tools can help in determining the difficulties that may affect people's independence [8]. For training, these tools can contribute to spatial orientation by improving correct way-finding behaviours [9]. Orientation difficulties usually occur due to impairments or diseases, such as stroke [10], Alzheimer's disease [11], acquired brain injury [12], and healthy aging [13]. Human spatial orientation in real life is based on both self-motion and visual cues (including proprioceptive and vestibular cues) [14]. Two technologies that can greatly assist in the development of tools for the assessment and training of spatial memory are Virtual Reality (VR) and Augmented Reality (AR). These two technologies can be of great help when the subject has to perform physical movement in the real world. The fact that the subjects have to move around the real world to perform the tasks resembles what they experience in real life. Moreover, physical displacement has been shown to be important in acquiring spatial ability skills [15].

This work presents a framework for the development of AR based on SLAM (Simultaneous Localization and Mapping). An AR mobile application was developed for this type of assessment involving visual and tactile stimuli by using our framework. Our AR application can be used in any indoor environment (e.g., a hospital ward, the therapist's office, or the patient's home) and requires the subject to physically walk around the real world. The task that the participants must perform is divided into two phases, a learning phase and an evaluation phase. In the learning phase, the participants physically walk around a room, and have to remember the location of virtual geometrical shapes. In the evaluation phase, the participants are asked to recall the locations where the shapes were in the room and place them correctly using AR. Our study compares short-term spatial memory involving visual (visual condition) and tactile (tactile condition) senses. A similar protocol was designed for the sense of touch, but using real objects. In the learning phase, the subjects touch real objects that are placed inside boxes in a real room, and they must remember their location. In the evaluation phase, the participants touch real objects that are hidden inside boxes and they have to place a virtual box that represents the touched object in the correct location. To our knowledge, this is the first work that uses AR to compare visual and tactile stimuli for the assessment of short-term spatial memory.

## 2. Assessment of spatial memory using virtual and augmented reality

Spatial memory is typically assessed in the visual modality, using paper and pencil tests [16, 17]. The development of computerized tools that use VR or AR provides some advantages over traditional tests. VR and AR applications allow objective indicators of the spatial learning of an individual (e.g., successes-failures, reaction times, speed, distance travelled, etc.) to be obtained and stored for later study. The presentation of stimuli can be varied and controlled [18–20]. Since VR and AR applications provide advantages regarding the evaluation and

training of people in a real environment (i.e., lower economic and time costs, etc.), the use of these technologies in the study of human ability has increased.

Different works have used VR to assess spatial memory in humans [21–24]. In the first VR applications for the assessment of spatial memory, the users were sitting in front of a computer screen without performing physical displacement. The users explored a VR environment during the task [18, 19, 25]. However, physical displacement is an important aspect to consider for spatial ability [15]. Some works have included physical displacement in VR environments [26–28]. Rodríguez-Andrés et al. [26] studied the influence of two different types of interaction on a VR task (an active physical condition versus an inactive physical condition). From their results, no statistically significant difference in task outcomes was found for the two types of interaction. They concluded that the type of interaction did not affect the children's performance in the VR task. Even though the physical movement was performed on the Wii balance board, the users did not have the feeling of walking in the real world. Cárdenas et al. [28] developed a maze-based VR task in which the participants had to perform physical movement. As in the previous work, they considered two different types of interaction (an active physical condition versus an inactive physical condition). The performance outcomes on their task were better for the inactive physical condition. As in the previous work, the physical movement was performed on a bicycle and the participants did not have the feeling of riding a bike in the real world, even when the user had to pedal on a stationary bike and had the HMD on.

Very few works have been presented for the study of the performance of spatial memory using AR. Our research group has presented most of these works. Our group has pioneered the development and validation of AR applications for the assessment of spatial memory [20, 29–31]. In the first work [20, 29], we developed an AR application that used image targets that were distributed in the real world. In the second work [30, 31], we presented an AR application based on SLAM for visual stimuli. The main difference with the previous work was that no additional elements were added to the real environment. The application used in this second work was created using the framework that is presented in this paper, but it only focuses on visual stimuli. Our group has also studied auditory stimuli for the assessment of spatial memory [32]. Meanwhile, Keil et al. [33] have recently presented a development and a study that are closely related to our work. They developed an AR system for location memory and distance estimation. The system shows holographic grids on the floor. They carried out a study to determine the effects of grids on the floor on location memory and distance estimation in an indoor environment. The results of their study showed that distance estimations were more accurate when a grid was displayed. The performance of location memory was worse when a grid was displayed.

Other works, which are not as closely related to ours, have focused on the use of AR to help in navigation. Chu et al. [34] presented a mobile AR tool to help in navigation. The scene captured by the camera was augmented with virtual 2D icons to guide users to their destination. Their study compared an AR tool and two map navigation modes. Their conclusion was that the AR tool was better than the two map navigation modes. Rehman and Cao [35] presented an AR application to help people navigate indoors. They compared the navigation using Google Glass, a smartphone, and a paper map. Their results showed that the participants perceived Google Glass to be the most accurate tool. No statistical differences were found between Google Glass and the smartphone in terms of workload. Google Glass and the smartphone were better than the paper map in terms of lower workload and less time of execution. However, the route retention was better with the paper map. Peleg et al. [36] presented a tool for route planning in public transportation. The performance using a mobile AR application and a non-AR application was compared. The augmented scene showed the expected times that buses go through each station on the map. They carried out a study in which they compared the

performance of older participants and younger participants. The participants who used the AR application completed the task in less time, but with higher error rates. These results were independent of the age of the participants. The younger participants showed significantly faster performance compared to the older participants while using the AR application. However, there were no significant differences regarding error rates.

## 3. Design and development

In this section, we describe the SLAM-based AR application used for the assessment of spatial short-term memory and how it has been used to configure the environment and in the visual condition. We describe the steps followed in the tactile condition and how the use of the AR application and touching real objects have been combined. We also present the three applications that were developed to automate tasks that have traditionally been evaluated using pen and paper. We finally present the hardware and software used for the development of our applications and the architecture of the framework presented in this paper.

### 3.1. Configuration of the environment

The AR application based on SLAM is designed to assess spatial short-term memory for shape location. The AR application runs on a smartphone. The participants have to perform a task that consists of walking around a real room. The dimensions of the room are 5 x 7.5 (around 38 m$^2$). The participants must look for virtual geometrical shapes and remember their locations. These geometrical shapes can only be seen through the mobile screen. For this purpose, the scene must be configured in two phases: room scanning and shape configuration.

The room scanning phase. In this configuration phase, the supervisor scans the real room. This scan stores the geometry of the room in the smartphone (Fig 1A and 1B). This scan includes the walls as well as tables, chairs, computers, etc.

The shape configuration phase. The supervisor places the different geometrical shapes in the real scanned room (Fig 1C) using the geometry stored in the smartphone. The application has about fifty 3D models of different types. In our study, we used eight geometrical shapes. These eight shapes are: cube, rectangular prism, wide cylinder, narrow cylinder, sphere, semi-sphere, cone, and pyramid. The application allows objects to be placed on planar surfaces (horizontal or vertical). In our case, they were all placed on tables. The bottom side of the geometric shape is attached to the detected surface. The shapes cannot be rotated. The shapes are always facing the camera position of the smartphone, and their up vectors are perpendicular to the plane of the surface.

### 3.2. The visual condition

For the visual condition, to use the AR application, the user must complete the following two phases. Fig 2 graphically shows the steps followed by the users in the visual condition.

The learning phase.   The main objective of this phase is to allow the subject to explore the room searching for virtual geometrical shapes and to remember their locations. The supervisor guides the users through the room. The users see the shapes in the same order and follow the same path. The established order, which was the most suitable for seeing all of the shapes in a continuous tour, was: cone, narrow cylinder, cube, pyramid, sphere, rectangular prism, wide cylinder, and semi-sphere. The participants have to indicate on the application that they have seen a shape by tapping on it. At that moment, a box appears to show that this shape has been marked as seen. Once the shape has been marked by the user, it cannot be seen anymore.

The evaluation phase.   This phase evaluates the participants' ability to remember the location of the virtual geometrical shapes in the real room. The users are asked to place all of the

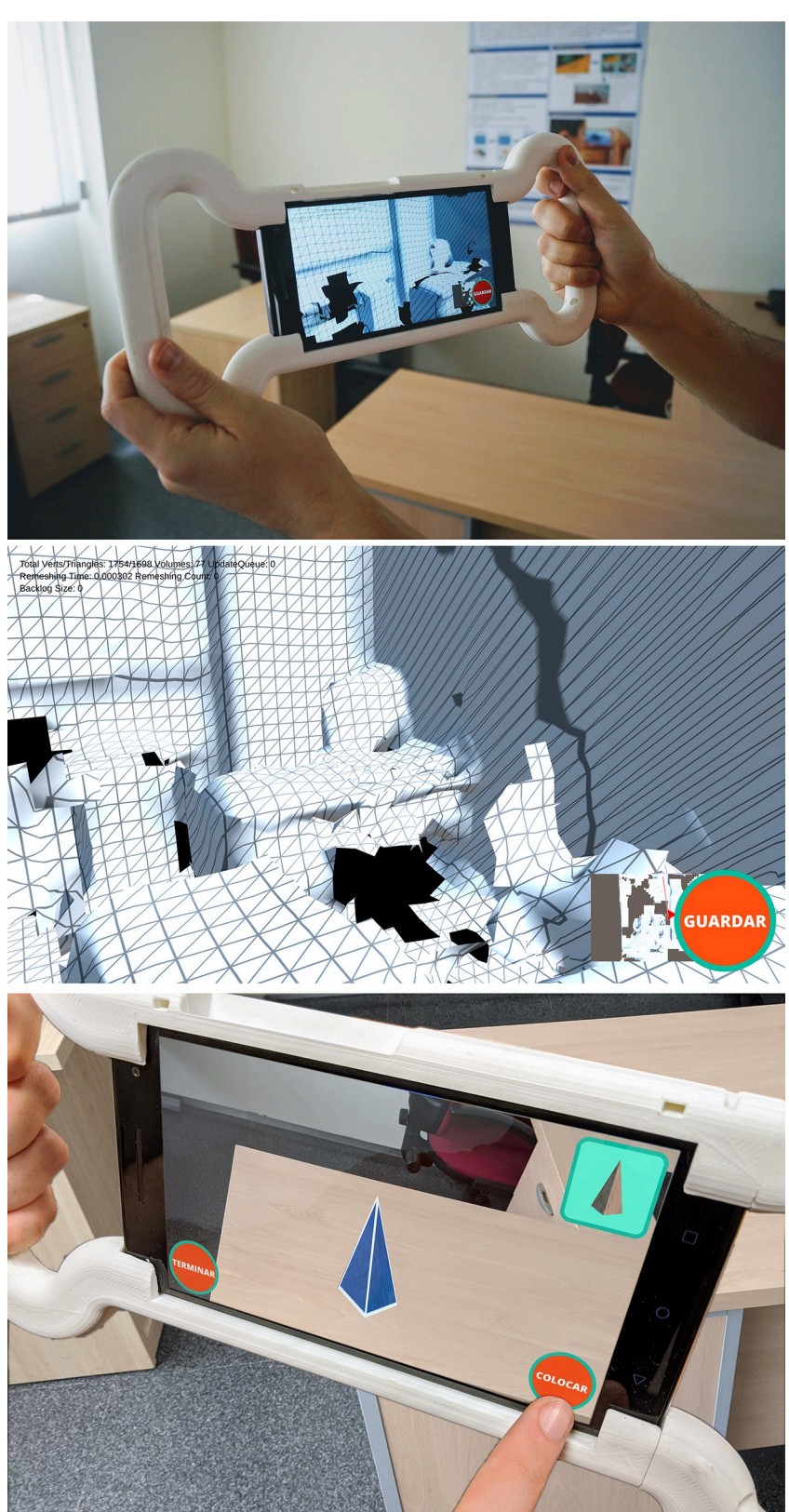

**Fig 1. Configuration phase.** (A) and (B) room scanning phase. (C) Shape configuration phase (an example of the supervisor placing a pyramid in the real scanned room).

eight shapes in their correct locations using the application. The shapes have a pre-established random order so that all of the participants carry out the placement in the same order. This order is different from the order followed in the learning phase: cube, rectangular prism, wide cylinder, narrow cylinder, sphere, semi-sphere, cone, and pyramid. The placement of the shapes does not have to be exact. There is a proximity tolerance. The point where the shape is placed by the user can be up to half a meter away from the exact point. The participants have

### Experiment room

### Virtual shapes

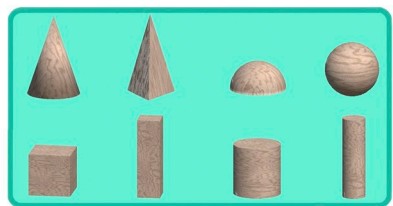

### LEARNING PHASE

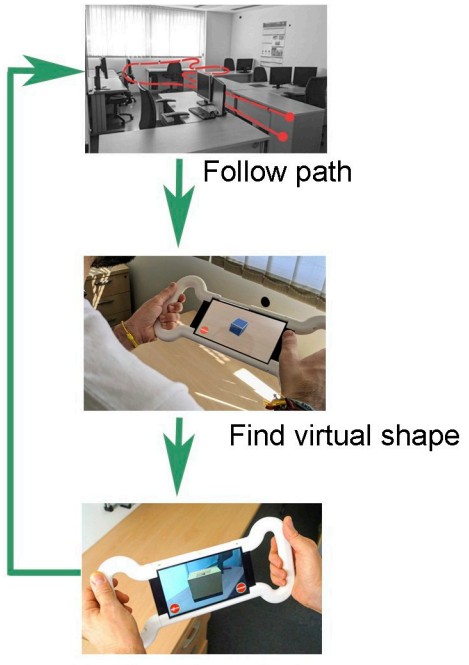

Follow path

Find virtual shape

Tap to acknowledge

### EVALUATION PHASE

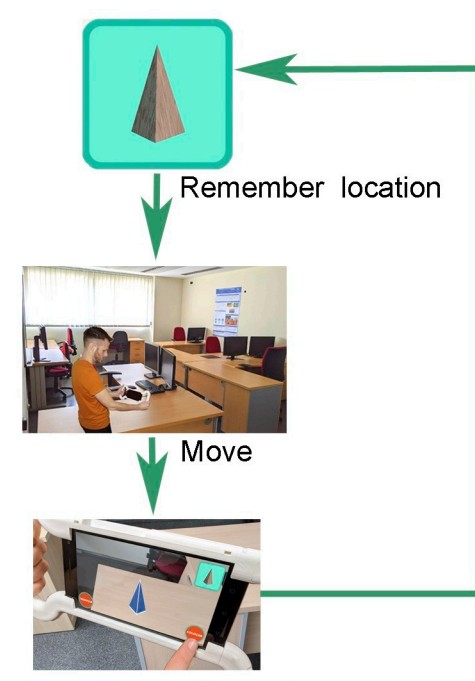

Remember location

Move

Place (three attempts)

**Fig 2. The steps followed by the users in the visual condition.** The study with the visual condition is performed in two phases: learning and evaluation. During the learning phase, the subject has to follow a predefined path, searching for the eight virtual shapes placed on the desks in the room. When a shape is found, the subject acknowledges it by tapping the screen. In the evaluation phase, subjects are requested to place each of the eight virtual shapes in its original location. Subjects have three attempts to place each shape.

three attempts to place each shape correctly. They are informed of whether or not they have placed the shape correctly in each try. On the last attempt, the shape remains fixed in the last position where it was placed, and the user is informed whether it was a success or failure. The participants are not helped regarding where the shapes are located.

### 3.3. The tactile condition

The task related to tactile stimuli combines the use of the AR application with touching real geometrical shapes inside opaque boxes. The geometrical shapes used are the ones already mentioned: cube, rectangular prism, wide cylinder, narrow cylinder, sphere, semi-sphere, cone, and pyramid (Fig 3A). The dimensions of the geometrical shapes are between 4 and 25 cm$^2$ at the base and between 3 and 10 cm in height. The boxes are closed. Their size is 31x22x25 cm with a capacity of 18 liters. The interior of the box is accessed through two small windows on the lower-left and lower-right sides, which are covered with strips of fabric. The subjects can introduce their two hands inside the box and touch the geometrical shapes, but they cannot see inside. Fig 3B shows the box. There are eight boxes. Each of the eight boxes had a different shape inside, and each box was identified using a barcode. The phases are the same as in the visual condition. Fig 4 graphically shows the steps followed by the users in the tactile condition.

**The learning phase.**   The boxes are placed in the room in the same locations as the virtual geometrical shapes were placed in the visual condition. In other words, the box containing the real sphere is placed in the same location where the virtual sphere was in the visual condition. The main goal of this phase is to allow the subject to walk around the room, to explore the real geometrical shapes by touch and to remember their locations. Since the real shapes are the same as the virtual ones used by the visual condition, they are located in the same locations. The subjects follow a pre-set tour, and they all perform the same task. When a subject reaches a box, he or she puts his/her hands in it and touches the shape to identify it and to remember its location. The AR application is used to scan the barcode and to store a timestamp of the moment that the shape was examined (Fig 3C).

**The evaluation phase.**   After finishing the learning phase, the subject is invited to leave the room for a moment. Meanwhile the supervisor removes every real box from its place and piles them altogether on a working table in a given order. The working table is another table that is not used in the learning phase. The subject comes back to the room again and stands in front of the working table. The supervisor takes a real box (working box) and puts it on the working table. The subject touches its shape by introducing his or her hands inside of the working box without seeing the shape. Using the AR application, the supervisor scans the barcode that is attached to the box being used, thereby informing the AR app of the shape inside. Then, the supervisor gives the AR application to the subject. The subject has to place a virtual box in the real position where he or she had touched that shape before. There are no physical boxes on the tables. The virtual boxes are only seen through the screen of the mobile device.

While the user is performing the placement of the box, the supervisor takes the current working box and puts it on a new pile. Then, the supervisor takes the next working box and puts it on the working table for the next evaluation step. When the user has placed the current box at the desired place, he or she returns to the working table with the AR device. The process is repeated for all eight boxes. The order is the same for all of the subjects. Fig 3D shows a subject placing a virtual box in the room.

### 3.4. The shape-recognition application

The shape-recognition application allows the participants to select the shapes memorized in the previous tasks from among different shapes. The participants are asked to identify the

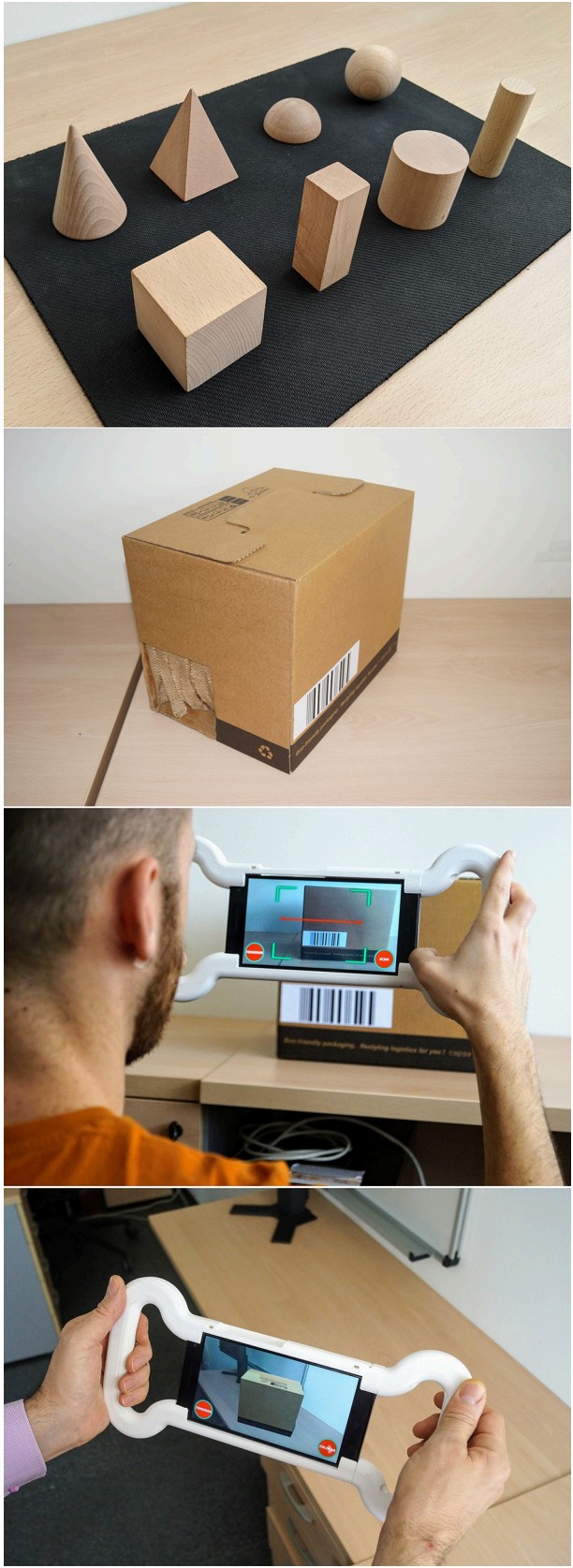

**Fig 3. Materials used in the tactile condition.** (A) Physical geometrical shapes used in our study. (B) Box used in the learning and evaluation phases of the tactile condition. (C) An example of the AR application for scanning the barcode. (D) An example of a participant placing a virtual box in the evaluation phase using the AR application.

geometrical shapes that they have touched or seen among a collection of different 3D shapes that appear on the screen of a tablet (Fig 5A). To facilitate the recognition of the shapes, they are animated and rotate around their vertical axes. The subjects can select and deselect as many shapes as they desire. Once the subjects have finished with the selection of shapes, they notify the supervisor, who ends the test. The application shows fifteen objects on the screen, of

### 8 boxes with shapes
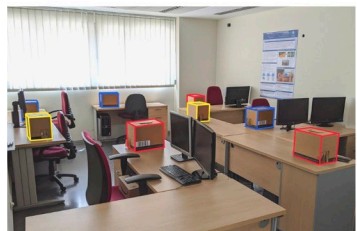

### Physical shapes
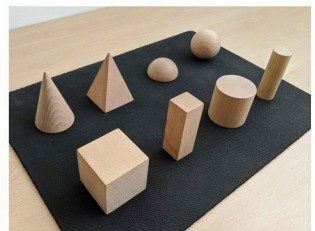

### Boxes are removed
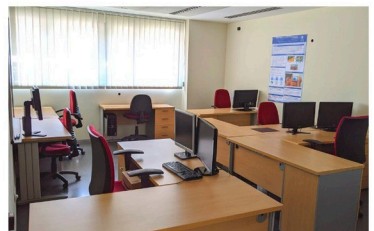

### LEARNING PHASE
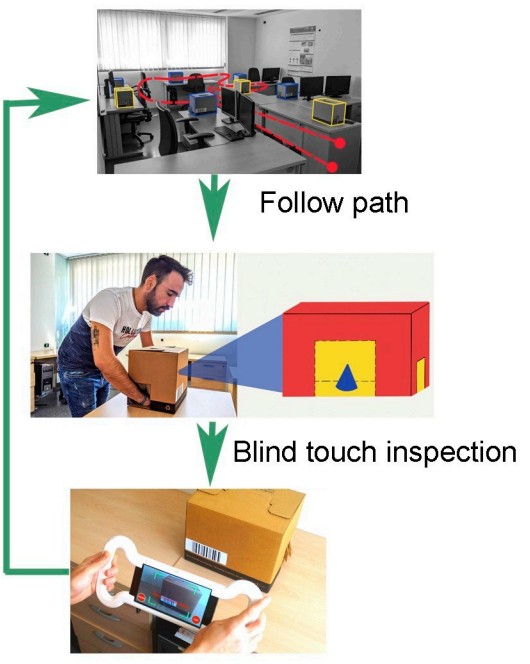

### EVALUATION PHASE
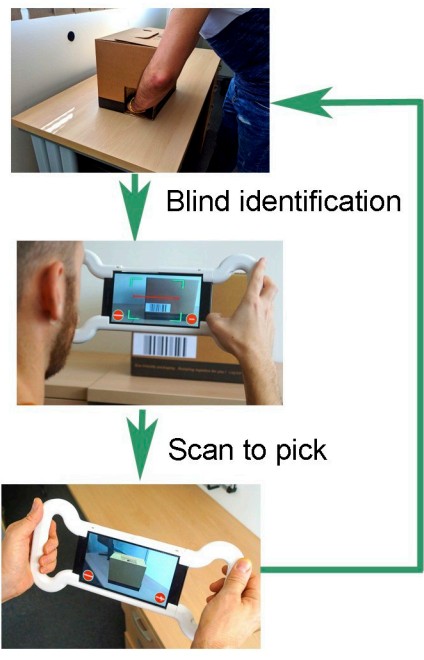

**Fig 4. The steps followed by the users in the tactile condition.** The study with the tactile condition is performed in two phases: learning and evaluation. During the learning phase, the subject has to follow a predefined path with eight physical boxes. Each box has a different geometrical shape inside. For each box, the subject has to inspect the shape inside by touch through two holes in the sides of the box. Then, the AR application is used to scan the barcode and to store a timestamp of the moment that the shape was examined. In the evaluation phase, the boxes are removed from their initial position and shown to the subject one by one. After inspecting its shape, the subject has to place a virtual box with the AR application in the position where the shape was located during the learning phase. The subjects have three attempts to place the virtual box correctly.

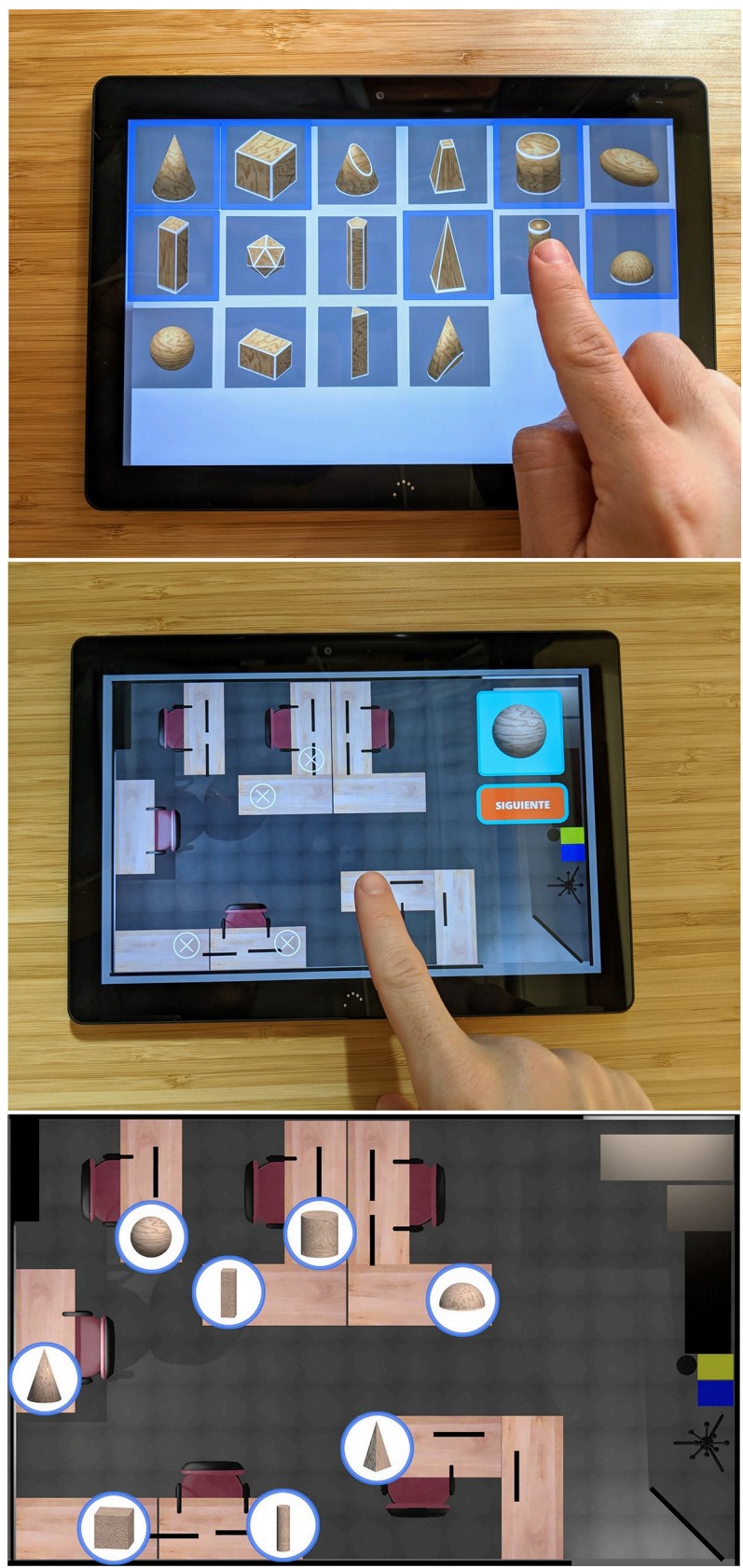

**Fig 5. Other applications and a schematic top view of the room used.** (A) Shape-recognition application for selecting geometrical shapes. (B) Map-pointing application used in the visual condition to place the geometrical shapes. (C) Schematic top view of the room and the location of the eight virtual geometrical shapes.

which our study uses eight objects. The order used to present the shapes on the screen is always the same.

## 3.5. The map-pointing application

The map-pointing application shows an empty map of the room. This map is a two-dimensional simplified map of the room in which tables, chairs, computers and office furniture are displayed. The participants must place the geometrical shapes that they have touched or seen in the right place. For the visual condition, the eight shapes are presented on the top right of the screen one at a time (Fig 5B). Once a given shape is presented to the subject, he or she has to place it on the map. For the tactile condition, the participants insert their hands into a box and touch a shape without seeing it. They place that shape (represented as a box) on the virtual 2D map using the tablet. The maximum distance between the position selected by the subject and the real position is also 0.5 meters (scaled to the map). The subjects have one chance to place the shape in the right place, and they are not notified whether or not they have placed it correctly. It is not possible to change the position of a shape that has already been placed.

## 3.6. Spatial orientation application

The spatial orientation application is a computer version of the PTSOT (Perspective Taking/ Spatial Orientation Test) test [37]. The PTSOT test is a traditional paper-and-pencil spatial orientation test. The PTSOT test consists of twelve items that evaluate the users' ability to orientate themselves spatially and think about different perspectives. The application shows one PTSOT image at a time. It presents a circle around a given object placed in the center. A vector that joins the center of the circle with a point on the circumference indicates the position of a different element. The user is asked for the related position of another object using a second vector. The user has to move this second vector with his/her finger to place it in the desired position. Once the user has set the vector in the desired position she or he can move to the next orientation question.

## 3.7. Hardware & software

All of the developments were developed using the Unity real-time development platform. Unity was chosen because it allows the integration of practically all of the AR SDKs. The device used for the development and study was a Lenovo Phab 2 Pro. The main features of the smartphone are: resolution (1440 x 2560 pixels); dimensions (179.8 x 88.6 x 10.7 mm); screen size (6.4 inches); and weight (259 grams). This smartphone can run applications developed with Tango SDK, implementing the SLAM technology, which is possible thanks to its three built-in cameras (a depth camera, a wide-angle camera, and a color camera with 16 MP). Another device that can be used to develop a similar version is the iPad Pro, which includes a LiDAR scanner.

A PC convertible was used for all tests in which the subjects had to be seated (the shape-recognition task, the map-pointing task, and the PTSOT test). This PC was a 13" HP Pavilion X360 with a resolution of 1366 x 768 pixels. To fill out the questionnaires, the participants used a 13" Macbook Pro laptop.

## 3.8. Architecture of our SLAM-based AR framework

To facilitate the development of the AR application used in this work and other different applications, we designed and implemented a SLAM-based AR framework. This section describes the functionalities that are specific to our framework and that have been explicitly developed for it. Our framework uses some specific services that the AR SDK used must provide; in this case, it is the Tango SDK, but it could be ARKit or any other. There are two specific services: recognition of an environment and positioning of the device in it. The framework facilitates access to these two services by providing an interface with specific functions that are common to all of the SDKs used. The developer abstracts from the different specific mechanics of each SDK. Moreover, the framework offers high-level AR functions that are not implemented in the AR SDKs such as: management of environments where AR is used, placement of virtual objects in the environment, generation of files to store the configuration of objects anchored to the environment, etc. The framework also offers functions outside the AR SDK, such as user management or data storage during sessions. Therefore, in the design of the architecture of our framework, two groups of functionalities were defined: functionalities related to AR technology and other functionalities.

The following functionalities were identified for AR technology:

- scanning of 3D geometry of real environments,

- storage, recovery, and management of 3D geometry in real environments,

- importation of 3D models to be used in the augmented environment,

- configuration of the scanned environments by adding the desired visual elements.

The following functionalities were identified for non-AR technology:

- management of user sessions,

- management of the times set for the different tasks,

- storage of the events produced by users during the task. The information stored includes direct interactions with the environment (e.g., the position of the device) or interactions through the interface (e.g., selected objects, pressed buttons, or placed objects).

Additionally, the following goals in the design were defined:

- building a modular architecture, which allows adding or removing features in a simple way. This modularity also facilitates team development since each team member can work on different modules.

- implementing the inter-module communication in a simple and efficient way. This communication should be as robust, efficient, and simple as possible.

- focusing on the AR-related functionalities. The framework must be able to work with different AR SDKs/engines.

The architecture of this framework was designed in three layers as shown in Fig 6.

- The core layer. This is the central layer of the framework. This layer is in charge of managing the subscription of the different modules to the framework. It controls the communication between the different modules and allows the scene layer to access the interfaces of the modules. This layer also manages the life cycle of the different framework components in order to ensure consistency in their updates and calls.

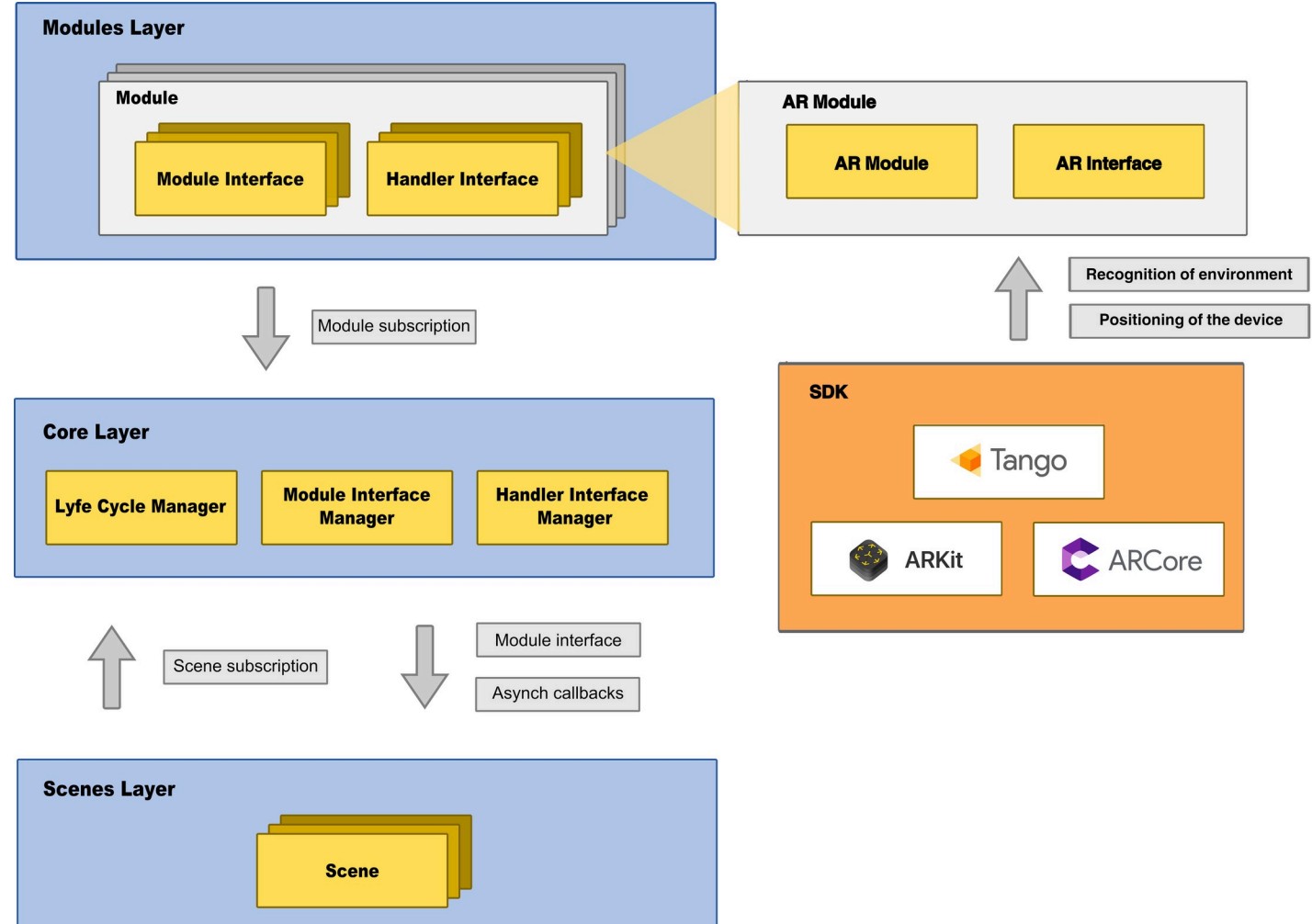

**Fig 6. Architecture of our SLAM-based AR framework.**

- The module layer. This layer consists of the different modules that are available in our framework. Each of these modules implements a different functionality that can be added to the applications. Each module must have at least one available interface (module interface) that can be called from the scene layer or from other modules. Handler interfaces, which manage the asynchronous outputs of the module, are also implemented.

- The scene layer. This layer contains the different scenes of the applications. Each scene has access to all of the modules. The scenes access the modules through the core layer. The scenes can also implement handler interfaces to receive asynchronous callbacks from modules.

This architecture is centralized. All of the elements involved in the framework subscribe to the core layer. All of the processes that occur in the framework go through the core layer. All of the updates are triggered by the core layer in an orderly way. This layer manages the order in which processes must be executed. In other words, no element can manage its own processes in isolation from the core. Therefore, there is no background computation that can

affect the performance of the application. The access to the modules is achieved through the core layer. In addition, this architecture is modular. It is distributed in independent modules that can communicate with other modules, but always through the core layer and by interfaces. The core layer manages the integration of the different modules. A module only has to subscribe to the core layer, indicating what its module interfaces and its handler interfaces are. The core layer manages all subscriptions and allows the scene layer to access those interfaces. The core layer is also in charge of managing the subscription of the scenes to the module callbacks once the interfaces are implemented.

Specific modules have been implemented for AR. These modules can be used with different AR engines/SDKs in a simple way and without altering the rest of the code. The core layer also has an event-handling system. The events can reach a multitude of components that are subscribed to the core layer. This communication system is useful for sending broadcast messages with different content. There are templates for different events with different objectives. There are broadcast messages regarding the interface to indicate, for example, which system component is allowed to receive interface events. There are also messages indicating the status of the AR service.

The AR application used in this study was implemented using our framework. There is a scene in the application for each of the phases of the task. These scenes subscribe to the core layer of the framework and can use the different modules. The modules used in this application are: the AR module, the user interface module, the configuration storage module, the event and data storage module, and the time control module.

The AR module is the module that is in charge of providing all of the AR functionality. This module encapsulates the initialization and configuration of the AR engine. It offers different simple methods that are common to different AR engines. Some of these functionalities are: scanning, storage, and management of environments where AR is used, detection of planes in the environment, placement of virtual objects in the environment, generation of files to store the configuration of objects anchored to the environment, etc. It also offers user interface events for the management of augmented objects. It should be noted that this module facilitates the switch among the different AR engines.

The user interface module manages everything that is displayed on screen as well as the interactions between the user and the application. This module is divided into two sub-modules: an AR interface sub-module that manages the part involving AR elements, and the standard interface module. The AR sub-module implements functions to manage the user's interaction with augmented objects in the environment and also with the current environment shown on the screen. The standard sub-module controls features that are common to all applications and allows the creation of menus, buttons, selection screens, etc.

The configuration storage module is in charge of storing and retrieving the configuration files of the scenes. The geometry of the environment, the selected objects and their respective locations in the environment are stored for each scanned environment. The module for storing events and related data is in charge of managing the identifiers of the session and the user. In addition, this module stores all of the information that is relevant to the tasks performed by the user, all of the user's interactions with the application, and the outcomes. It also stores the physical paths that users have taken during their tour in the real environment. The time control module manages the time specified for each task and provides configuration options. It also allows interface elements (e.g., countdown timer) to be incorporated.

Three more applications were developed to carry out the study presented in this paper: the shape-recognition application, the map-pointing application, and the PTSOT application. These three applications were developed with a model-view-controller architecture, using the user interface tools offered by Unity. The storage of outcomes, events, times, etc. was carried out using the same storage structure as defined above.

# 4. Study

## 4.1. Participants

The initial sample included 53 adults (60% undergraduates and 40% graduates). The gender distribution was 70% men. The participants were recruited at the Universitat Politècnica de València (Spain), through campus advertising. The participants received a small reward consisting of a USB stick. The final sample was selected after applying two inclusion criteria that are explained in section 4.2. The final sample considered for our study included 47 adults. The gender distribution was 70% men. The minimum age was 20 and the maximum age was 56. The mean (standard deviation) age of the participants was 30.18 (9.25) for males, and 32.86 (11.22) for females. Two of the participants were left-handed. None of the participants were under medical treatment that could affect their cognitive functioning. None had suffered a brain injury, or had any sensory or motor impairment. None of the participants were familiar with the environment where the study was carried out. The participants gave written informed consent prior to the study. The individuals who appear in the images of this manuscript gave written informed consent to publish their case details. The study was approved by the Ethics Committee of the Universitat Politècnica de València, Spain. The study was conducted in accordance with the declaration of Helsinki.

## 4.2. Statistical tests

The Shapiro-Wilk test was used to check the normal distribution of the variables [38]. None of the variables followed a normal distribution. Therefore, non-parametric tests were used with the whole dataset. A descriptor of each group is presented in the format (median (Mdn); interquartile range (IQR)). All of the tests are presented in the format (statistic *U/W*, normal approximation *Z*, *p-value*, *r* effect size). The results were considered to be statistically significant if p < .05. We used the R open source statistical toolkit (https://www.r-project.org) to analyze the data (specifically, RStudio 1.2.5033 for Windows).

## 4.3. Inclusion criteria

We built a battery of haptic tests to assess tactile abilities. These tests were used as inclusion criteria to select participants with similar tactile abilities. Our battery of tests is based on the battery of haptic tests (Haptic-2D) proposed by Mazella et al. [39]. Their complete battery consists of eleven haptic tests divided into five categories to measure short-term memory, scanning skills, spatial comprehension skills, tactile discrimination skills, and picture comprehension. Two-dimensional raised stimuli (lines, dots, patterns, shapes, or pictures) were printed on swell paper for their tests. They used a wooden structure to enable subjects to introduce their hand behind the curtain to explore the stimuli through active touch, thus preventing subjects from seeing the materials. Our battery of tests consisted of tests for spatial orientation, shape discrimination, and spatial location built in plastic and in swell paper. Our plastic tests were printed on a Rapman 3.1. 3D printer that used an additive process, where successive layers of material are laid down according to the desired pattern. Our 3D printer used ABS white material. The items were printed in pieces of two different sizes, 26 x 6 cm and 30 x 7 cm. We also used swell paper to raise stimuli and create items equivalent to those used in the plastic items. The size of the paper items was the same as the size of the printed items, 26 x 6 cm and 30 x 7 cm. To prevent the users from seeing the tests, we built a structure of cardboard with a thickness of 5 mm and an open area with a curtain made of fabric so that the users could introduce their hands, but not see inside (Fig 7A). The three tests using the plastic items are described below.

1. Spatial orientation test. The stimuli in the spatial orientation test are raised-line figures made up of one, two, or three rectilinear segments. Each segment has a specific orientation (vertical, horizontal, or oblique). Each segment is 3 cm long. Each test includes a practice item and six items with increasing complexity. Our test includes two items with one segment, two items with two segments, and two items with three segments. The subjects must explore the figures with the index finger of their dominant hand. We used the instructions proposed by Mazella et al. [39]. Each item consists of a reference element and four similar elements. The participants must inspect the reference element by touch, and then touch the following four elements one by one, without being able to go back. For each of the four elements, the participants must indicate verbally whether or not the figure has the same spatial orientation as the reference. The supervisor writes down their answers without giving any feedback. Two points are awarded for each correctly completed item. The maximum score is 12 points.

2. Shape discrimination test. The protocol is similar to the spatial orientation test. The only difference is the type of figures, which are geometrical shapes. The size of the shapes is adjusted so that they occupy the maximum size in an area of 9 cm$^2$. These 2D shapes are: square, rectangle, rhombus, sphere, semi-sphere, ellipse, pentagon, hexagon, right triangle, equilateral triangle, and five-pointed star. Fig 7B shows an item used in this test. As in the previous test, the participants must inspect the reference element by touch. Then, for each of the four elements, the participants must indicate verbally whether or not the shape of the figure is the same as the reference element.

3. Spatial location test. The protocol is similar to the two previous tests. The only difference is the type of figures. The stimuli in this test are raised-line figures inside a circle (diameter = 4 cm). The circle may contain one, two, or three small elements inside (circle, star, or square). The size of these small elements varies from 7 to 10 mm. Three of the items have one inner element, two items have two inner elements, and one item has three inner elements. As in previous tests, the participants must inspect the reference element by touch. Then, for each of the four elements, the participants must indicate verbally whether or not the spatial location of the inner elements is the same spatial location as in the reference element.

The swell paper test contains a practice item and six items (Fig 7C). This test includes two items for spatial orientation, two items for shape discrimination, and two items for spatial location. The protocol is similar to the three previous tests. However, there are two differences: the type of material used to build the test, and the total number of items (6).

After building a box plot with the scores obtained by the participants in all of the haptic tests, five outliers were identified. These five outliers were removed from the sample (N = 48). We checked whether or not there were differences for the haptic test battery and for gender. To determine whether or not there were differences for the total score of the three plastic tests of our battery between the group of women (Mdn = 28; IQR = 3.5) and the group of men (Mdn = 28; IQR = 2), we applied the Mann-Whitney U test (U = 200, Z = -.740, p = .467, r = .108). To determine whether there were differences for the total score of the swell paper test score between the group of women (Mdn = 10; IQR = 0) and the group of men (Mdn = 10; IQR = 4), we applied the Mann-Whitney U test (U = 246.5, Z = .381, p = .712, r = .056). These results indicate that the women and men of our sample had the same tactile abilities.

We checked the total score of the participants in the PTSOT test as an inclusion criterion (Fig 7D). We used the previously published PTSOT scores as the upper limit for men and women [30]. These upper limits are 57.61˚ for men and 92.03˚ for women. One male was excluded from the sample (N = 47). We checked whether or not there were differences for the

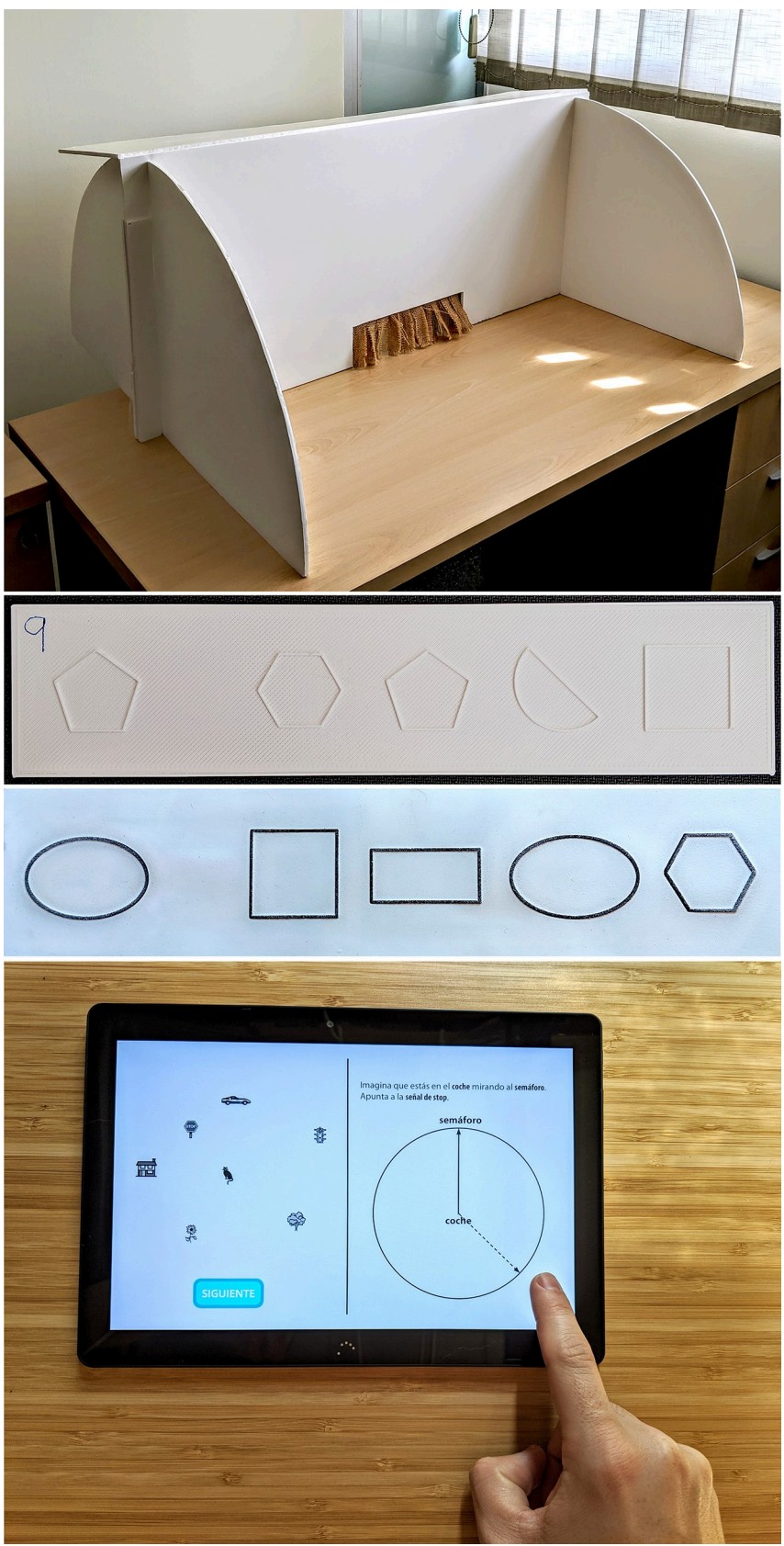

**Fig 7. Tests used as inclusion criteria.** (A) Structure to prevent the participant from seeing the items used in the battery of haptic tests. (B) An example of one item included in the shape discrimination test (printed using ABS white material). (C) An example of one shape discrimination item included in the swell paper test. (D) An example of a user interacting with the PTSOT application.

PTSOT variables and for gender. To determine whether or not there were differences for the PTSOT score between the group of women (12.65; 10.555) and the group of men (12.86; 11.56), we applied the Mann-Whitney U test (U = 212, Z = -.442, p = .671, r = .064). To analyze the unattempted items on the PTSOT for the group of women (33.333; 20.833) and the group of men (33.333; 16.667), we applied the Mann-Whitney U test (U = 243, Z = .284, p = .785, r = .041). These results indicate that the women and men of our sample had similar spatial orientation abilities.

## 4.4. Measures

**Performance with visual and tactile conditions.** The visual application and the tactile application store the following variables: the total number of shapes located correctly during the evaluation phase (LocShapes); the total number of attempts made while placing the shape in the correct location (AttemptS); and the total time required to complete the learning phase (TTimeL) and the evaluation phase (TTimeE).

**The shape-recognition task.** We developed the shape-recognition application to test the ability of the participants to recognize the shapes that they inspected during the task. The shape-recognition task stores two variables: the number of shapes that the participants selected correctly (RecogS), and the number of shapes that the participant selected incorrectly (RecogI).

**The map-pointing task.** We developed the map-pointing application to test the ability of the participants to read a bi-dimensional map of the room in which the main task with the visual/tactile application occurred. The map-pointing application stored one variable: the number of shapes that the participant placed correctly on the map (MapShapes). A shape is considered to be correctly located on the map when the participant places it within a radius of what would be half a meter in the real world, which is the equivalent of 3 cm on the virtual map.

**Questionnaires.** The participants filled out a questionnaire about their subjective experience with the AR application. The questionnaire consists of 21 questions that we group into the following variables: enjoyment, concentration, usability, competence, calmness, expertise, non-mental effort, non-physical effort, presence, and satisfaction. This questionnaire was designed specifically for this study, and some of the questions were adapted from commonly used questionnaires [40–43] and based on our previous experiences [31, 44]. The items were on a 7-point Likert scale, ranging from 1 "Totally disagree" to 7 "Totally agree". All of the items were formulated in a positive manner except for items #9 and #10. Scores for the variables were obtained by calculating the mean value of the associated questions. This questionnaire is available in the S1 Appendix.

## 4.5. Procedure

The study was carried out in a room of 37.5 square meters. The room had the furniture commonly found in an office or in a university laboratory. Fig 5C shows a schematic top view of the room and the location of the eight virtual geometrical shapes.

The participants were involved in the two conditions (tactile and visual). Both conditions were tested in sessions that were held more than two months apart. After more than two months, it was very difficult for the participants to remember the location of the shapes.

Furthermore, the room was not familiar to the participants since they had not had previous access before the study. The two conditions were the following:

- The tactile condition: participants who learn the location of the physical geometrical shapes that were placed inside of physical boxes distributed throughout the room. In the learning phase, the boxes were placed in the room as shown in Fig 3D. The participants had to touch the shapes inside the boxes. The application was used to store which shapes were touched and when. In the evaluation phase, the boxes were not distributed throughout the room. The boxes were all on a table. The users had to touch the shapes one by one, in the same order. After touching a physical geometrical shape, the users were asked to place a virtual box using the AR application in the location that they thought they had touched the real object in the learning phase.

- The visual condition: participants who learn the location of the virtual shapes placed in the real room using our AR application. The participants used the AR application for the learning and evaluation phases. The location of the geometrical shapes (real or virtual) was the same in both conditions (Fig 5C).

The administration protocol was comprised of two sessions. In the first session, the participants completed the battery of haptic tests and the PTSOT test. Then, they performed the tasks in the tactile condition: the short-term memory task for the location of geometrical shapes, the shape-recognition task using the shape-recognition application, and the map-pointing task using the map-pointing application. After more than two months, the participants completed the tasks again using the visual condition: the short-term memory task with the AR application, the shape-recognition task, and the map-pointing task. Finally, the participants filled out an online questionnaire about their subjective perception regarding the use of the AR application.

## 5. Results

### 5.1. Performance outcomes

We compared the performance outcomes between the two conditions (TactileCondition vs. VisualCondition) (within-subjects analysis) to determine how the use of tactile or visual stimuli affects memory of the location of stimuli. Fig 8 shows box plots for the performance outcome variables and for the tactile and visual conditions. First, we analyzed the variable that indicates the total number of successes for the eight stimuli used. This variable counts the number of shapes placed correctly (up to eight shapes). To determine whether or not there were differences in remembering and placing shapes in their correct locations between TactileCondition (Mdn = 7; IQR = 1) and VisualCondition (Mdn = 8; IQR = 1), we applied the Wilcoxon Signed-rank test ($W = 123$, $Z = -1.739$, $p = .098$, $r = .179$). This result indicates that there are no statistically significant differences between the two conditions.

Second, we analyzed the variable that represents the total number of attempts made when placing the shape in the correct location. Note that a maximum of three attempts were allowed to place a shape in the correct location. To determine whether or not there were differences for this variable between TactileCondition (Mdn = 5; IQR = 4) and VisualCondition (Mdn = 3; IQR = 5), we applied the Wilcoxon Signed-rank test ($W = 492$, $Z = 2.860$, $p = .004$, $r = .295$). This result indicates that there are significant differences between the two conditions in favor of the visual condition, which required fewer attempts to place the shape correctly.

Third, we analyzed the variable that represents the total time required to complete the learning phase. To determine whether or not there were differences for this variable between TactileCondition (Mdn = 123.54; IQR = 44.64) and VisualCondition (Mdn = 128.02;

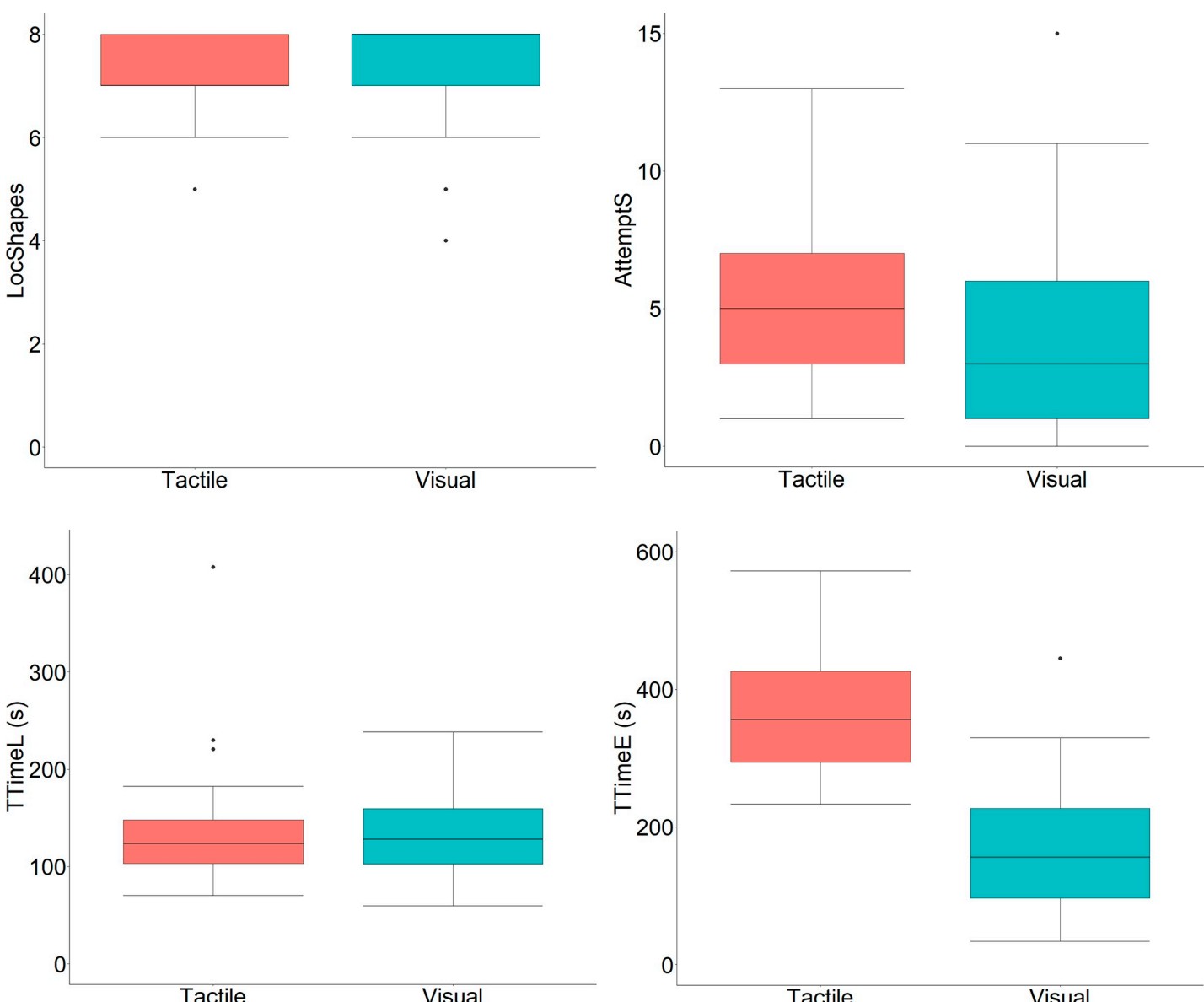

**Fig 8. Box plots for the performance outcome variables and for the tactile and visual conditions.** (A) Total number of shapes placed correctly (LocShapes variable). (B) Total number of attempts required to correctly place the shapes (AttemptS variable). (C) Total time in seconds required to complete the learning phase (TTimeL variable). (D) Total time in seconds required to complete the evaluation phase (TTimeE variable).

IQR = 56.84), we applied the Wilcoxon Signed-rank test ($W = 454.5$, $Z = -1.159$, $p = .249$, $r = .120$). This result indicates that there are no significant differences between the two conditions.

Fourth, we analyzed the variable that represents the total time required to complete the evaluation phase. To determine whether or not there were differences for this variable between TactileCondition (Mdn = 356.23; IQR = 132.325) and VisualCondition (Mdn = 156.11; IQR = 130.44), we applied the Wilcoxon Signed-rank test ($W = 1127$, $Z = 5.958$, $p < .001$, $r = .614$). This result indicates that there are significant differences between the two conditions in favor of the group with visual stimuli, who spent less time to complete the evaluation phase.

We also analyzed the variable of the shape-recognition test that represents the number of shapes that the participant remembers after performing the task. To determine whether there

were differences for this variable between TactileCondition (Mdn = 8; IQR = 0) and Visual-Condition (Mdn = 8; IQR = 0), we applied the Mann-Whitney U test ($W = 28$, $Z = -1.387$, $p = .178$, $r = .143$). This result indicates that there are no significant differences between the two conditions. We also analyzed the variable that represents the number of shapes that the participant selected incorrectly for TactileCondition (Mdn = 0; IQR = 0) and VisualCondition (Mdn = 0; IQR = 0), and we applied the Wilcoxon Signed-rank test ($W = 15$, $Z = 2.236$, $p = .037$, $r = .231$). This result indicates that there are significant differences between the two conditions in favor of the group with visual stimuli, which selected fewer incorrect shapes.

We also analyzed the variable that represents the number of shapes that the participant placed correctly in the map-pointing task. To determine whether there were differences for this variable between TactileCondition (Mdn = 4; IQR = 3) and VisualCondition (Mdn = 6; IQR = 1), we applied the Mann-Whitney U test ($W = 127$, $Z = -4.250$, $p < .001$, $r = .438$). This result indicates that there are significant differences between the two conditions in favor of the group with visual stimuli, which placed more shapes correctly.

## 5.2. Gender analysis and subjective perception

To determine if gender influences the variable that indicates the number of correctly selected shapes, we analyzed the group of women (Mdn = 8; IQR = 1) and the group of men (Mdn = 7; IQR = 2) for TactileCondition, and we applied the Mann-Whitney U test ($U = 289$, $Z = 1.438$, $p = .154$, $r = .210$). For VisualCondition in the group of women (Mdn = 8; IQR = 1) and the group of men (Mdn = 8; IQR = 1), we also applied the Mann-Whitney U test ($U = 221$, $Z = -.262$, $p = .803$, $r = .038$).

We then analyzed the total number of attempts. For TactileCondition in the group of women (Mdn = 4; IQR = 3.5) and the group of men (Mdn = 5; IQR = 4), we applied the Mann-Whitney U test ($U = 151$, $Z = -1.874$, $p = .063$, $r = .273$). For VisualCondition in the group of women (Mdn = 3; IQR = 5) and the group of men (Mdn = 3; IQR = 6), we applied the Mann-Whitney U test ($U = 235$, $Z = .094$, $p = .934$, $r = .014$).

We analyzed the time required to complete the evaluation phase. For TactileCondition in the group of women (Mdn = 343.915; IQR = 64.173) and the group of men (Mdn = 395.23; IQR = 162), we applied the Mann-Whitney U test ($U = 203$, $Z = -.651$, $p = .527$, $r = .095$). For VisualCondition in the group of women (Mdn = 166.255; IQR = 105.653) and the group of men (Mdn = 123.83; IQR = 132.58), we applied the Mann-Whitney U test ($U = 252$, $Z = .489$, $p = .637$, $r = .071$).

Since no statistically significant differences were found in any of these analyses, we can conclude that the performance results were independent of the participants' gender.

The questionnaire about subjective experience was used to measure the participants' subjective perception of the AR application and their performance in the Visual Condition. The questions were grouped in the following variables: enjoyment, concentration, usability, competence, calmness, expertise, non-mental effort, non-physical effort, satisfaction, and presence. No statistically significant differences were found in any of these variables; therefore, we can conclude that the subjective perception was independent of the participants' gender ($U \geq 169.5$, $Z \geq -1.513$, $p \geq .134$).

## 5.3. Correlations

This section presents the significant correlations obtained after the application of the Spearman rank correlation and the Bonferroni correction for the visual condition. For the tactile condition, no significant correlations were found between the AR spatial task and other tasks.

We used the Spearman rank correlation to test the associations among variables for the performance in the visual condition (LocShapes, AttemptS) and among variables in the shape-

recognition task (RecogS) and the map-pointing task (MapShapes). As there are multiple comparisons, the Bonferroni correction was applied. If we apply the Bonferroni correction to the matrix formed by these four variables, LocShapes correlates with MapShapes ($r = .49$, $p < .01$), and AttemptS also correlates with MapShapes ($r = -.48$, $p < .01$). This sensibly means that participants who placed fewer shapes correctly during the evaluation phase or required more attempts to place the shapes correctly also made more mistakes on the map-pointing task. Fig 9A and 9B show these two correlations graphically.

If we apply the Bonferroni correction to the matrix formed by the four variables (TTimeL, TTimeE, LocShapes, and AttemptS), the TTimeE correlates with the LocShapes variable ($r = -.51$, $p < .01$) and with the AttemptS variable ($r = .48$, $p < .01$). This means that participants who required more time during the evaluation phase placed fewer shapes correctly and required more attempts. Fig 9C and 9D show these two correlations graphically.

We also used the Spearman rank correlation to test the associations among the ten subjective variables (enjoyment, concentration, etc.) and the variables of the performance outcomes (LocShapes and AttemptS). If we apply the Bonferroni correction to the matrix formed by these 12 variables, the LocShapes variable correlates with perceived competence ($r = .49$, $p = .03$), and the AttemptS variable correlates negatively with perceived competence ($r = -.50$, $p = .03$). This means that participants who placed a greater number of shapes correctly felt greater competence. In contrast, the greater the number of attempts, the less the perceived competence. Fig 9E and 9F show these two correlations graphically.

If we apply the Bonferroni correction to the matrix formed by the ten subjective variables, there is only a significant correlation that corresponds to the enjoyment and presence variables ($r = .47$, $p = .04$). This means that participants who experienced more enjoyment reported a higher sense of presence. Fig 9G shows this correlation graphically.

## 6. Discussion

In studies related to working or short-term spatial memory, mainly visual or auditory modalities have been considered [20, 26, 29, 32, 33]. In this work, we contribute with a study in which the tactile modality is considered. To our knowledge, this is the first study in which tactile stimuli are used for the assessment of short-term spatial memory for shape-location in a three-dimensional array. In our study, the participants navigated the environment to learn the spatial location of each tactile stimulus.

There are substantial differences in terms of the studies carried out in our previous AR works and the study presented in this paper. In the study of Munoz-Montoya et al. [31], only four objects were used and the participants were divided into two groups in which the learning phase was different (AR vs. Non-AR). The participants who used AR learned the placement of the virtual elements in the real world using AR. The participants who used Non-AR learned the location of the virtual elements through photographs. The augmented photographs showed the real environment with the virtual objects superimposed on it. In the study of Munoz-Montoya et al. [30], eight virtual objects were used (horse, wall clock, telephone, toy car, bust, fountain pen, sailboat, and screwdriver). The objects were distributed in a building on two different floors. That study analyzed the correlations between the results of the performance outcomes with the AR app taking into account gender and different measures of emotional and spatial factors. These measures were the levels of trait anxiety, wayfinding anxiety, and different spatial strategies for wayfinding.

From our results, there was no significant difference for the number of shapes placed correctly between the two conditions (visual vs. tactile). However, the participants in the visual condition required significantly less time to complete the evaluation phase and also required

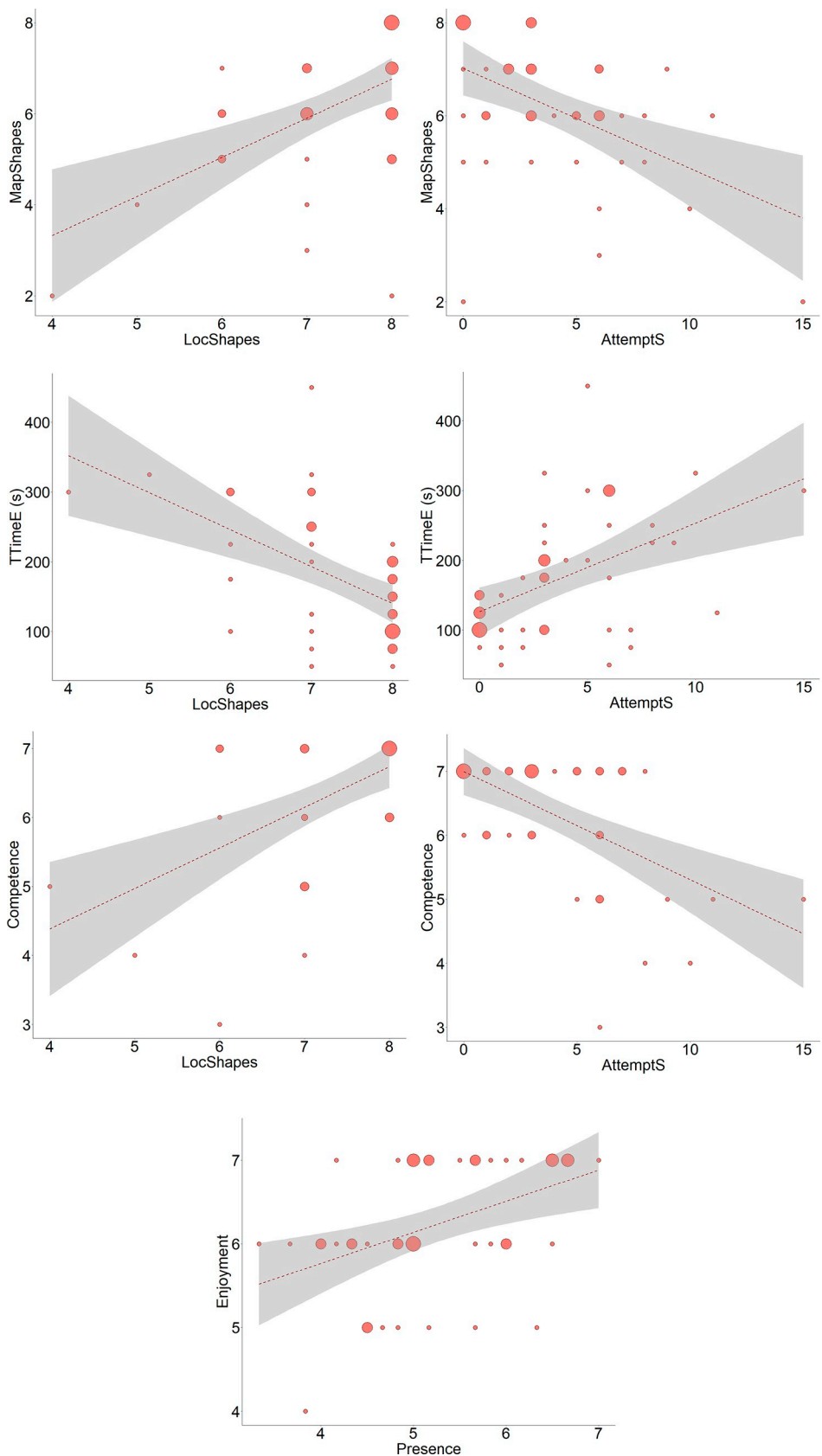

**Fig 9. Scatter plots for significant correlations for the visual condition (n = 47).** (A) Scatter plot for the total number of shapes placed correctly (LocShapes variable) and the score obtained in the map-pointing task (MapShapes variable). (B) Scatter plot for the total number of attempts to place the shapes correctly (AttemptS variable) and the score obtained in the map-pointing task (MapShapes variable). (C) Scatter plot for the total number of shapes placed correctly (LocShapes variable) and the time spent in the evaluation phase (TTimeE). (D) Scatter plot for the total number of attempts to place the shapes correctly (AttemptS variable) and the time spent in the evaluation phase (TTimeE). (E) Scatter plot for the total number of shapes placed correctly (LocShapes variable) and the perceived competence variable. (F) Scatter plot for the total number of attempts to place the shapes correctly (AttemptS variable) and the perceived competence variable. (G) Scatter plot for the presence and enjoyment variables. Three sizes of circles appear in the plots that represent a low number of occurrences, an intermediate number of occurrences, and a high number of occurrences. The dashed red lines are best fitting linear regression lines. In panels C, D and G close values were grouped for clearer visualization. The grey area represents a 95% confidence level interval for predictions from a linear model.

significantly fewer attempts. There was a significant difference in the number of shapes that the participants placed correctly in the map-pointing task after performing the task in favour of the visual condition. There was no significant difference for the shape-recognition test, which represents the number of shapes that the participants remember after performing the task between the two conditions (visual vs. tactile). However, the variable that represents the number of shapes that the participants selected incorrectly showed that the participants in the visual condition selected fewer incorrect shapes. From all of these results, we can conclude that visual and tactile stimuli can be used to assess spatial memory and that visual stimuli positively affects short-term, spatial memory outcomes. This supports previous research stating that the sense of sight is the dominant sense in humans, and, consequently, this dominance facilitates the learning of spatial-visual associations [45, 46]. Our results also indicate that visual and tactile stimuli are suitable for the assessment of spatial memory in navigation regardless of the gender of the participants. These results are in line with previous works [20].

The following correlations were identified between the total number of shapes correctly remembered and located using visual and tactile stimuli (LocShapes) and the total number of shapes correctly placed in the pointing-map task (MapShapes). For the Visual Condition, LocShapes correlated with MapShapes. This result shows that spatial-visual associations were learned and transferred from the three-dimensional array of the real room to the bi-dimensional array of the map. In contrast, spatial-tactile associations were learned, but failed to achieve this transfer. These results are in line with previous works, in which the memory for the location of virtual objects in the visual modality in a navigational space correlated positively with the pointing of these objects on a map [30].

We also studied the correlations between the subjective perception and the performance measures of the participants for the AR spatial task (Visual Condition). The total number of shapes (LocShapes) correlated with perceived competence, which indicates that the greater the number of correctly placed shapes, the greater the perceived competence. The AttemptS variable (attempts required for placing the shapes) correlated negatively with perceived competence. This correlation indicates that the more attempts required, the less competence was perceived.

For the AR application, the subjective perception of the participants was independent of their gender. On a scale from 1 to 7, the medians were very high: equal to or above 6 in all cases (except for one with a value of 5.3). The correlations among the subjective variables were analyzed for the whole sample, and thirteen positive correlations were found. It is worth highlighting that the sense of presence correlated with the enjoyment variable. The participants that reported a higher sense of presence also experienced more enjoyment.

The developed framework allows different AR engines/SDKs to be used. For example, these different AR engines could be ARCore or ARKit, which are wrapped in a module with the same interface offering, and therefore have the same functionalities. This design allows the rest of the

application to be independent of specific AR SDKs. To take into account that different AR engines may have different functionalities, we have defined a set of basic functionalities. Another set of optional functionalities was also defined, which can be activated or deactivated depending on the engine used. We have tried to minimize this last set of functionalities so that the impact on the rest of the application code is minimized. At this time, the Tango SDK could be changed for another equivalent engine without altering the application code. Until the release of ARKit 3.5 in March 2020, neither ARKit nor ARCore offered enough functionalities to afford the development of applications like the one presented in this work. ARKit 3.5 incorporates the possibility of using the new LiDAR scanner and depth-sensing system that are built into the iPad Pro. Thanks to these new sensors, it is possible to use the geometry of the scene, just as is done with Tango SDK. Therefore, applications such as those presented in this work can be created by using ARKit in supported devices. Tango SDK has been deprecated since March, 2018. Since then, with the discontinuation of compatible devices on the market (Lenovo Phab 2 Pro and ASUS Zenfone AR), there has been a gap that was not covered. Therefore, the appearance of the iPad Pro is especially relevant because it opens up many possibilities for developers.

Our proposal and other similar tools can greatly help in training short-term spatial memory. Difficulties in spatial memory are usually associated to disorders or diseases (e.g., [47, 48]). For example, several studies have reported that hippocampal damage may provoke impairments in short-term memory (e.g., Koen et al. [47]). Patients with amnestic mild cognitive impairment have difficulty remembering where the objects are located (e.g., Hampstead et al. [48]). In an ongoing work, patients with acquired brain injury perform a task with visual stimuli with our AR application. From the preliminary results, we have observed that these patients can perform the task with the AR application. Therefore, our AR application could be used as a help tool for groups with impairments in short-term memory.

This work studies the differences in seeing and touching objects and recognizing their shapes by touch. To our knowledge, no work has been presented in which a haptic system allows the recording of user responses related to a navigational task for the learning of spatial-tactile associations. However, different commercial haptic devices or prototypes developed by researchers have been presented for their use in VR and help users to have a higher level of presence. Dangxiao et al. [49] classified the paradigm of haptic HCI into three stages (desktop haptics, surface haptics, and wearable haptics). They presented a taxonomy of desktop haptic devices and also a classification of tactile feedback devices. One of the most widely used wearable haptic devices are haptic gloves. These gloves can include hand motion tracking and provide distributed force and tactile feedback in the palm and in the fingertips. They compared 10 existing commercial haptic feedback gloves. Existing commercial VR controllers (e.g., HTC Vive) only provide global vibrotactile feedback, which greatly limits the haptic experience. The existing haptic technology does not allow a user to touch a virtual object and recognize its shape. An open challenge is how to efficiently include localized and diverse spatial-temporal vibrotactile patterns, thermal feedback, texture feedback, contact, softness, skin tightening, etc. in a haptic device [49]. In the study carried out in this work, the participants touched real objects, and the use of haptic devices is proposed as future work.

We built a haptic test battery to assess tactile abilities. We used this battery as an inclusion criterion in order to assure that the participants involved in our study had similar tactile abilities. We checked whether or not there were differences in the haptic test battery (plastic and swell paper) and for gender. The results indicated that the haptic test battery was independent of the participants' gender.

We developed applications for the tasks and tests that are traditionally done on paper. Specifically, we developed the shape-recognition application, the map-pointing-application, and the PTSOT application. The supervisor's work is facilitated by being able to perform these

tasks or tests on the computer. The applications themselves control the times and also store successes and errors. The applications are particularly useful in making precise calculations (e.g., the angles for the PTSOT test). One of the advantages of using software tools is that the process of data analysis is facilitated [50]. In a paper-based approach, the data must be digitized for analysis, which can be time consuming depending on the amount of data to be digitized. When using software tools, the data only has to be exported to spreadsheets. While the paper-based approach can take hours, using software tools is a matter of minutes. Our applications automatically create the spreadsheets. Other authors have argued that the replacement of traditional paper-based neuropsychological tests by computerized neuropsychological tests offers other advantages since the traditional tests are expensive and time consuming [51]. It might be interesting to complement the information about the recognition memory task with a free-recall task. However, in our case, we used geometrical shapes and many participants would have had problems correctly describing the eight shapes they had to memorize. Therefore, we decided to evaluate the ability of the users to identify the eight memorized shapes with a recognition task involving seven additional shapes in order to monitor the strength of the memory formed about the shapes.

A limitation of our study is that the two sessions were carried out in the same order (first the tactile and then the visual). The design of our study could have influenced the better performance in the vision session due to participants remembering shape locations from the previous session. Nevertheless, we would like to highlight that the participants carried out the second session more than two months later, the type of objects used were not remarkable to induce a long-term memory, and the room in which the study was carried out was not familiar to the participants. These three aspects lead us to argue that it would be very difficult for a participant to remember the specific location of a certain geometric shape in the room. However, this limitation could have been removed if the order of the sessions were counterbalanced.

## 7. Conclusions

This paper introduces the architecture of our framework for the development of SLAM-based AR applications. Our framework allows the use of different AR engines/SDKs and allows the development of applications that do not depend on these engines/SDKs. An AR application for the assessment of short-term spatial memory was developed. Our application works in any indoor environment and can even be used in several rooms or on several floors of the same building. The supervisor can configure the environment and include as many virtual shapes as desired and place the shapes in any position of a real space.

For the first time, we have carried out a study in which tactile stimuli are used for the assessment of short-term spatial memory in a navigational space. A SLAM-based AR application was used to compare visual and tactile stimuli. From the results, we can conclude that visual and tactile stimuli can be used to assess spatial memory. The number of shapes placed correctly was similar for both conditions. The tactile condition required more time and more attempts to complete the task. The performance outcomes were independent of gender, but a tendency of women requiring fewer attempts than men was found for the tactile condition. Therefore, the tactile stimuli are stimuli that can be used to assess the ability to memorize spatial-tactile associations. However, more research is needed to further investigate how the sense of touch and other sensory modalities (excluding visual and auditory) can be used to assess spatial memory and identify the groups that can benefit from them.

## Supporting information

**S1 Appendix.**
(DOCX)

**S2 Appendix.**
(CSV)

## Acknowledgments

We would like to thank all of the people who participated in the study. We would like to thank the editor and reviewers for their valuable suggestions.

## Author Contributions

**Conceptualization:** Francisco Munoz-Montoya, M.-Carmen Juan, Magdalena Mendez-Lopez, Ramon Molla, Francisco Abad, Camino Fidalgo.

**Formal analysis:** Francisco Munoz-Montoya, M.-Carmen Juan, Magdalena Mendez-Lopez.

**Funding acquisition:** M.-Carmen Juan, Magdalena Mendez-Lopez.

**Investigation:** Francisco Munoz-Montoya.

**Methodology:** Francisco Munoz-Montoya, M.-Carmen Juan, Magdalena Mendez-Lopez, Ramon Molla, Francisco Abad, Camino Fidalgo.

**Software:** M.-Carmen Juan, Ramon Molla, Francisco Abad.

**Supervision:** M.-Carmen Juan.

**Writing – original draft:** Francisco Munoz-Montoya, M.-Carmen Juan, Magdalena Mendez-Lopez.

**Writing – review & editing:** Francisco Munoz-Montoya, M.-Carmen Juan, Magdalena Mendez-Lopez, Ramon Molla, Francisco Abad, Camino Fidalgo.

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
