## [Decision Letter · Decision Letter 0]

5 Oct 2020

PONE-D-20-20120

A SLAM-based augmented reality framework for the assessment of short-term spatial memory. A comparative study of visual versus tactile stimuli

PLOS ONE

Dear Dr. Juan,

Thank you for submitting your manuscript to PLOS ONE. After careful consideration, we feel that it has merit but does not fully meet PLOS ONE’s publication criteria as it currently stands. Therefore, we invite you to submit a revised version of the manuscript that addresses the points raised during the review process.

I apologize for the delay in reaching a decision on your manuscript. Originally, two expert reviewers had agreed to evaluate your manuscript. Unfortunately, only one reviewer was able to complete their assessment of your work. Instead of seeking a third reviewer’s opinion, to avoid further delays I decided to review your manuscript myself. Both Reviewer 1 and myself find the manuscript interesting and think it may merit to be published. Before I can fully evaluate this however, there are several issues that need to be addressed. These issues pertain the clarity of the manuscript, the rigor of the analyses, and the presentation of the results. Please respond to all the comments raised by Reviewer 1 and myself, clearly highlighting the changes made to the manuscript to address each issue. I look forward to receiving your revised manuscript.

We look forward to receiving your revised manuscript.

Kind regards,

Guido Maiello

Academic Editor

PLOS ONE

Additional Editor Comments:

Major Comments

1) Sections 1.1 and 2.2 in the Introduction are overly lengthy. All of the literature discussed is indeed relevant and should be mentioned. However, I suggest the authors summarize the main findings of the studies they mention, reduce this whole section to a couple of paragraphs, and move most of the content of these two sections to the Discussion section of the manuscript. Even for the portions of these section that will be moved to the Discussion, I would strongly suggest that many details should be removed altogether. For example, there is no need to specify the number of participants in each study, or the details of experimental conditions. The authors should simply summarize in one sentence the findings of each study and comment on how their current work relates to this previous literature.

2) Page 19 line 513: “This guaranteed that the participants could not remember the location of the shapes.” This claim is unsubstantiated. It is well possible that participants at least partly remembered the location of objects from one session to the next. The fact that the vision session always occurred after the haptic session is thus a confound. This means that we cannot completely exclude that the better performance in the vision session was due to participants remembering shape locations from the previous session. Ideally, the study should be repeated with new participants, and by randomizing object location and counterbalancing session order. I strongly suggest the authors adopt these practices in future studies. For the current study, I imagine collecting new data would be challenging due to the covid-19 pandemic. Therefore, at the very least this limitation of the study should be clearly noted in the discussion section of the manuscript.

3) Results need to be presented more clearly and rigorously.

a. To begin with, it would be useful to have plots showing the differences between haptic and vision sessions for each of the performance outcomes. Presenting medians and IQRs in parentheses scattered throughout the results makes it very difficult to understand and interpret the findings.

b. Asterisk notations should be kept consistent across the manuscript. In most places ** means p<0.05, but in Table 1 *p < .05 and ** p< .01. I would suggest removing asterisks from the main test, and only showing them in figures and tables, employing them in the standard notation: *p < .05; ** p< .01; ** p< .001

c. Page 23 line 622. This sentence is unclear and I do not understand what correlations were actually assessed.

d. Even though I do not fully understand exactly which correlations were tested, I believe the authors must have performed at least 60 correlation tests. Some correction for multiple comparisons needs to be employed. All non-significant results should not be discussed. For example, the authors should remove from page 22 line 601, and from the abstract, the mention of a “tendency” for gender difference. The authors should also remove lines 609 to 612, since the results for men and women appear to be the same.

e. It would also be useful to see scatter plots for all of the significant correlations the authors describe.

4) All of the data underlying the results of the manuscript needs to be made available. I would suggest the authors upload all their data and relevant analysis code to a public data repository such as Zenodo. If the authors do not wish to upload their data to a public repository before the manuscript is accepted, they should at least upload this data as supplementary material, or provide a private link where I can assess whether all the data is presented in a format that allows other researchers to reproduce the results presented in the study. It would also be useful and good research practice if the authors uploaded the software they describe in the manuscript (i.e. the various “apps”) to a public repository as well.

Minor Comments

Page 2 Line 39: “Most of the information that humans explicitly store in spatial memory comes from the visual and auditory modalities.” Please substantiate this claim with appropriate references.

Page 2 Line 55: “In contrast, physical displacement has been shown to be important in spatial ability” This sentence is unclear, I do not understand the point the authors are trying to make. Please rephrase/elaborate.

Page 11 line 293: “PTSOT” please define this acronym the first time it appears in the text.

Page 16: Please specify the tasks of the Shape discrimination and Spatial location tests. What were the participants reporting?

Page 17: the meaning of the numbers in parentheses - e.g. line 446: “(28; 3.5)” - is unclear, since this is explained only on page 20.

Page 17 line 473: “One men” should be “One man”, or “One male”

Page 22 line 608 and line 624: “p > ” should be “p <”

Page 25 line 678 “On a scale from 1 to 7, the medians were very high: equal to or above 6 in all cases (except for one with a value of 5.3).” This statement is unclear. The reader does not know what the questions on the questionnaire were, so one cannot judge what “high medians” represent.

Journal Requirements:

2. We note that Figure 2 includes an image of a participant in the study. 

Reviewers' comments:

Reviewer's Responses to Questions

**Comments to the Author**

1. Is the manuscript technically sound, and do the data support the conclusions?

Reviewer #1: Yes

2. Has the statistical analysis been performed appropriately and rigorously? 

Reviewer #1: I Don't Know

3. Have the authors made all data underlying the findings in their manuscript fully available?

Reviewer #1: No

4. Is the manuscript presented in an intelligible fashion and written in standard English?

Reviewer #1: No

5. Review Comments to the Author

Reviewer #1: The paper describes an AR framework for the assessment of short-term spatial memory. The presented work has a good basis of prior work, already published by the same authors. The differences and the novelty of the present work have been outlined and described by the authors. Nevertheless, I am not sure that PLOS one is the right journal for a work that I would define as "incremental". Without considering this issue, I think that the work is valuable, interesting, and supported by good results. Results seems complete and well presented.

In the following some issues I have encountered in the paper and that should be fixed.

1) Language. In some points authors used a sort of colloquial language. As an example, in the abstract they wrote "This paper presents a framework for the development of SLAM-based Augmented Reality apps for the assessment of spatial memory". What is the meaning of apps? Have you developed a framework (as in the title), a software, a mobile application? You should better define the software contribution of the work.

2) The previous point also has a consequence in the rest of the paper. In general, it is difficult for the reader to understand the real contribution of the paper. You described the framework/software/app but it is difficult for the reader to understand how it works. You should add more images showing the app working in the different cases, and better some link to videos showing the app working in the different modalities. You should better explain how the app works in the tactile situation.

3) Again about the choice of the terms, I am not convinced about the name "visual app", to define the AR software that shows virtual object in AR. As far as I understood the "visual app" is a "standard" AR app based on SLAM, so a less confusing name should be used to refer to it.

4) When describing the software, the same level of details has been provided for new features and for already implemented features (like the scanning if the room). You should better explain what is from the standard SDK you used and what you have developed.

5) Section 4 “Study” should be carefully checked. There are repetitions (see “The gender distribution was 70% men.} that is repeated twice), and numbers are given in a non-uniform way.

6) Questionnaires: since you used both standard questions and ad-hoc ones, you should provide the list of questions as they have been used at least in an appendix or as an external link.

7) Discussion. The works strongly rely on Tango SDK, which has been deprecated. You discussed it, but it is a big issue for the further development and use of the system you developed.

7b) Discussion: there is a big gap between discussion of sw related issues and discussion about other aspects, like the memory impairments (see lines 699-700).

6. PLOS authors have the option to publish the peer review history of their article (what does this mean?). If published, this will include your full peer review and any attached files.

Reviewer #1: No

---

## [Author Response · Author response to Decision Letter 0]

3 Nov 2020

A rebuttal letter that responds to each point raised by the academic editor and the reviewer was uploaded as a separate file labeled 'Response to Reviewers'.

---

## [Decision Letter · Decision Letter 1]

8 Dec 2020

PONE-D-20-20120R1

SLAM-based augmented reality for the assessment of short-term spatial memory. A comparative study of visual versus tactile stimuli

PLOS ONE

Dear Dr. Juan,

Thank you for submitting your manuscript to PLOS ONE. After careful consideration, we feel that it has merit but does not fully meet PLOS ONE’s publication criteria as it currently stands. Therefore, we invite you to submit a revised version of the manuscript that addresses the points raised during the review process.

Both Reviewer 1 and myself were pleased with how you addressed all the comments we raised. I have only a few remaining minor changes I would ask you to make, again to aid the clarity of the presented figures and results. Note that Reviewer 1 also commented on the quality of the figure resolution, so I recommend you double check the PlosOne figure formatting guidelines and make sure you upload all figure files at the right size, resolution, and with visible text. The remaining edits to the manuscript should not take much effort, and I don’t think there will be need for further rounds of revisions after this. I look forward to receiving your revised manuscript. 

We look forward to receiving your revised manuscript.

Kind regards,

Guido Maiello

Academic Editor

PLOS ONE

Additional Editor Comments (if provided):

• There is a sentence typo on line 117: “have recently presented a development and study that…”

• Section 5.3 should probably not be a section on its own, but can be incorporated in the previous section 5.2, since you are still evaluating gender effects.

• In Section 5.4, it would be useful if you explained in simple terms what the reported correlations mean. For example, you say:

“LocShapes correlates with MapShapes (r = .49, p < .01), and AttemptS also correlates with MapShapes (r = -.48, p < .01).”.

You could elaborate specifying that:

“This sensibly means that participants who placed fewer shapes correctly during the evaluation phase or required more attempts to place the shapes correctly also made more mistakes on the map pointing task.”

I know you elaborate on these results in the discussion section, but it’s difficult to keep in mind what all the variables here mean. Thus very simple explanations would be useful for all the correlations reported.

• I appreciate the new result figures; they greatly aid comprehension. These could still be improved in a few minor ways.

o Text size should be increased to improve readability.

o In all panels of figures 8 and 9 I suggest you scale the axes so they cover the full range of possible values starting from 0.

o Where appropriate, please specify axes units, for example in Figure 8b, the yaxis label could be “TTimeL (s) ”.

o To reduce clutter, only one legend is sufficient in Figure 8, since all subpanels have the same color scheme.

o In Figure 9, you could make all the axes square. More importantly, some dots in the scatterplots actually represent multiple data points. To show this, you could scale dot size by the number of occurrences for each data point. For an example see Figure 4 from https://doi.org/10.1080/07370024.2016.1243478

• In the results, it looks like you run correlation analyses only for the visual condition (lines 638-639). However, in the discussion section (line 718) you say that “In contrast, spatial-tactile associations were learned, but failed to achieve this transfer.” Does this mean you performed the same correlation analyses for the tactile condition and found no significant results? If so, please specify this. This probably means you simply have to add the previously-deleted sentence in the results: “For the Tactile Condition, no significant correlations were found between the AR spatial task and other tasks.”

Reviewers' comments:

Reviewer's Responses to Questions

**Comments to the Author**

1. If the authors have adequately addressed your comments raised in a previous round of review and you feel that this manuscript is now acceptable for publication, you may indicate that here to bypass the “Comments to the Author” section, enter your conflict of interest statement in the “Confidential to Editor” section, and submit your "Accept" recommendation.

Reviewer #1: All comments have been addressed

2. Is the manuscript technically sound, and do the data support the conclusions?

Reviewer #1: Yes

3. Has the statistical analysis been performed appropriately and rigorously? 

Reviewer #1: Yes

4. Have the authors made all data underlying the findings in their manuscript fully available?

Reviewer #1: Yes

5. Is the manuscript presented in an intelligible fashion and written in standard English?

Reviewer #1: Yes

6. Review Comments to the Author

Reviewer #1: The authors addressed all my comments from the previous revision. Tha paper has been improved, and the main issues are now solved.

In particular, I appreciated the new figures that now better explain the setup, the structure of the application and how it has been used for the experiments.

It is now clear that the use of the Tango framework is not a limitation, since it is clear that the approach can be extended to other frameworks, like ARKit or ARCore (though they are very different in the used techniques, and thus in performance).

I still have one comment about the quality of the plots: I see them at a very low resolution in both visualizations. This is expecially evident in plots where I cannot read labels very well.

7. PLOS authors have the option to publish the peer review history of their article (what does this mean?). If published, this will include your full peer review and any attached files.

Reviewer #1: **Yes: **Manuela Chessa

---

## [Author Response · Author response to Decision Letter 1]

4 Jan 2021

A rebuttal letter that responds to each point raised by the academic editor and the reviewer was uploaded as a separate file labeled 'Response to Reviewers_R2'.

---

## [Editor Report · Decision Letter 2]

12 Jan 2021

SLAM-based augmented reality for the assessment of short-term spatial memory. A comparative study of visual versus tactile stimuli

PONE-D-20-20120R2

Dear Dr. Juan,

We’re pleased to inform you that your manuscript has been judged scientifically suitable for publication and will be formally accepted for publication once it meets all outstanding technical requirements.

Kind regards,

Guido Maiello

Academic Editor

PLOS ONE
---

## [Editor Report · Acceptance letter]

22 Jan 2021

PONE-D-20-20120R2 

SLAM-based augmented reality for the assessment of short-term spatial memory. A comparative study of visual versus tactile stimuli 

Dear Dr. Juan:

I'm pleased to inform you that your manuscript has been deemed suitable for publication in PLOS ONE. Congratulations! Your manuscript is now with our production department. 

Kind regards, 

on behalf of

Dr. Guido Maiello 

Academic Editor

PLOS ONE